# Anxiety and depressive symptoms among home isolated patients with COVID-19: A cross-sectional study from Province One, Nepal

**Pratik Khanal**[1]☯*, **Kiran Paudel**[1,2]☯, **Suresh Mehata**[3], **Astha Thapa**[3], **Ramesh Bhatta**[4], **Hari Krishna Bhattarai**[5,6]

1 Institute of Medicine, Tribhuvan University, Maharajgunj, Kathmandu, Nepal, 2 Nepal Health Frontiers, Tokha-5, Kathmandu, Nepal, 3 Ministry of Health, Province One, Biratnagar, Nepal, 4 Asian College for Advanced Studies, Purbanchal University, Satdobato, Lalitpur, Nepal, 5 Nepal Development Society, Chitwan, Nepal, 6 Program in Global Health, Humanitarian Aid and Disaster Medicine, Università del Piemonte Orientale, Novara, Italy and Vrije Universiteit Brussel, Brussels, Belgium

☯ These authors contributed equally to this work.
* pratikkhanal@iom.edu.np

**Data Availability Statement:** All data generated in the study are included in the manuscript.

## Abstract

Home isolated patients infected with COVID-19 might be at increased risk of developing mental health problems. The study aimed to identify the prevalence and factors associated with anxiety and depression among COVID-19 home isolated patients in Province One, Nepal. This was a cross-sectional study conducted between February 17, 2021, to April 9, 2021. A total of 372 home isolated patients from Province One were phone interviewed in the study. Anxiety and depression were measured using a 14-items Hospital Anxiety and Depression Scale (HADS). Multivariable logistic regression analysis was done to determine the risk factors of anxiety and depression. Among home isolated COVID-19 infected participants, 74.2% and 79% had symptoms of anxiety (borderline: 48.7% and abnormal: 25.5%) and depression (borderline: 52.7% and abnormal: 26.3%), respectively. Watching television was significantly associated with lower odds of experiencing symptoms of anxiety and depression. Females had significantly higher odds of having depression symptoms compared to males while ever married, those with COVID-19 related complications, and those taking medicine for the treatment of COVID-19 symptoms had a higher likelihood of exhibiting symptoms of anxiety. A focus on improving the mental health well-being of COVID-19 infected patients in home settings with connection to the health services is warranted with timely psychological interventions.

## Introduction

The rapid emergence of Coronavirus-19 disease (COVID-19) is altering the psychology and interpersonal connections of millions of people all over the world. The initial emotional response to a pandemic is dread and uncertainty, which can be overpowering and can rise to

**Funding:** The authors received no specific funding for this work.

**Competing interests:** The authors have declared that no competing interests exist.

negative feelings such as tension, worry, and despair, which can lead to societal unrest and mental disease [1].

Several psychological, social, and economic problems have emerged as a result of the COVID-19 pandemic, such as loneliness, social isolation, anxiety, stress, depression, and fear of COVID-19 infection, loss of loved ones, gender-based violence, substance abuse, and job loss [2–4]. According to a nationwide survey in China, 34.1% of those who were quarantined or isolated had experienced at least one of the following mental health issues: acute stress, anxiety, depression, or sleep disorders [5], with the likelihood being higher in frontline workers, people with pre-existing mental health issues, and people with chronic physical health disorders.

Quarantine and isolation measures, which are being implemented quickly to control the health emergency, may have detrimental psychological and social consequences, particularly for those who are most vulnerable, such as frontline medical staff, children, and the elderly [6–8]. The majority of the foreseeable direct effects of quarantine and associated social and physical isolation, such as financial insecurity, boredom, irritation, feeling burdened, loneliness, and fear, are risk factors for mental health issues such as anxiety, depression, suicide, and self-harm [6,9] Several cross-sectional studies have found a significant prevalence of psychological distress symptoms among the general population during COVID-19. In Saudi Arabia, the prevalence of depression and anxiety was 17.1% and 10% [10] while the United Kingdom showed 22.1% and 21.6% of people suffering from depression and anxiety, respectively [11]. Prior studies have revealed that quarantine methods employed in the outbreaks of the severe acute respiratory syndrome (SARS) and the H1N1 influenza pandemic have also been attributed to a higher risk of psychological consequences [12–14] and this could have long term consequences [6,15].

The first COVID-19 case in Nepal was reported in a traveler from Wuhan on January 23, 2020 [16] while the first COVID-19 related death was reported in a post-partum women on 16 May 2020 [17]. During the initial phase of the COVID-19 pandemic, institution-based isolation measures were introduced by the Government of Nepal with dedicated COVID-19 hospitals and isolation facilities established at primary health facilities [18]. In the initial days of the outbreak, the Government of Nepal admitted all the positive cases irrespective of symptoms. Later, due to the overwhelming patient flow, a guideline for home isolation was in place which allowed asymptomatic and those with mild COVID-19 symptoms to stay at home with close supervision from local health workers [19]. While home isolation measures reduced strain on the health system, it posed a challenge in following up with the clinical status and patients felt difficulty in reaching the health facilities in case of difficulty [20]. Due to the limited health resources and fear of COVID-19, the mental health status of the home isolated patients affected by the COVID-19 pandemic could have been compromised. In our literature review, no previous studies have documented the mental health status of home isolated patients during the COVID-19 pandemic. A better understanding of the psychosocial problems of home isolated patients can provide important guidance in carrying out timely psychological interventions during the COVID-19 pandemic and in any future outbreaks. In this context, this study aimed to determine the prevalence and major influencing factors of anxiety and depressive symptoms among COVID-19 home isolated patients.

## Materials and methods

### Study design and study participants

A cross-sectional study was conducted via phone interview among COVID-19 infected patients who had spent at least 5 days at home isolation. During the time of the survey, an

isolation of 14 days was recommended for patients infected with COVID-19. The study participants were from Province One, Nepal. Province One is one of the seven provinces of the country and has 14 districts. It was the second most affected province in terms of number of COVID-19 infected cases after Bagmati province [21]. Among the 14 districts, Morang, Sunsari and Jhapa were the most affected districts-all plain districts bordering to India [21]. During the data collection period (February-April 2021), Nepal experienced an increase in COVID-19 cases due to the second wave, hospitals ran out of beds and oxygen along with other essential supplies was in critical shortage [22]. As hospitals were overwhelmed with the patients, those with mild or no symptoms were suggested to stay at home isolation. Data were collected between February 17, 2021, and April 9, 2021.

## Sampling strategy and data collection methods

The sampling frame of COVID-19 home isolated patients was available from the database developed by the Ministry of Social Development of Province One, Nepal. The database included the RT-PCR test positive cases whose test was performed between April 17, 2020, to January 23, 2021. RT-PCR test positive cases reported during the period were 30316 among which contact detail was available of 11543 COVID-infected patients. The sample size was calculated using the formula of a cross-sectional survey. The total sample size required was 372 considering the prevalence of anxiety at 41%, a margin of error of 5% and a level of confidence at 95%. Since the study was not conducted in this population, we took the prevalence from a previous study done among health workers in Nepal [23]. A simple random sampling technique using the RAND command in Microsoft Excel was used to select the participants from the sampling frame of 11543 COVID-infected patients. A total of 372 participants were recruited for the study. Those who were hospitalized or unable to speak or were not available or were unwilling to participate were excluded from the study.

Enumerators with previous data collection experience and academic background in public health were recruited and trained by the study team. All confirmed home isolated patients were contacted by telephone with an invitation to join the study. Those providing consent were interviewed using a structured questionnaire at their convenient time.

## Data collection measures

The anxiety and depression status of the participants were assessed using the 14-item Hospital Anxiety and Depression Scale (HADS). The HADS is a commonly used tool for measuring anxiety and depression in different settings in many countries including Nepal [23–26]. The Nepali version of the HADS has satisfactory psychometric properties with construct validity achieved for both sub-scales of anxiety and depression [27]. It has seven items each for measurement of anxiety and depression with each item scoring from 0 to 3, and the total score ranging from 0 to 21. The total scores of these tools were interpreted as normal (0–7), borderline abnormal (8–10) and abnormal (11–21). For further analysis, a score of more than 7 was considered as the presence of anxiety and depression [28].

The socio-demographic section of the questionnaire consisted of information on the participant's sex (male, female), age in years, education (No schooling, secondary, higher secondary, and graduation or above), family type (Nuclear and Joint/Extended), Occupation (employee and unemployed), and marital status (Ever married and never married). COVID-19 and behavioral related characteristics consisted of information of participants having health workers in family members, presence of COVID-19 symptoms, presence of comorbidity, complication during COVID-19, taking medicine for COVID-19 symptoms, use of the internet during isolation, watching TV during isolation, use of social media for COVID-19 information, use of

Ministry of Health and Population (MOHP)/WHO site for COVID-19 information, smoking and alcohol history.

## Data analysis

Descriptive analysis was done by calculating frequency and percentages for categorical variables. The Chi-square test was used to determine the association between categorical variables. To determine potential factors associated with the outcome variable, a multivariable logistic regression analysis was performed, and adjusted odds ratio (AOR) and 95% confidence interval (CI) were calculated. For adjusted regression analysis, those variables which were significant at a 10% significance level in bivariate analysis were included in the multivariable logistic regression analysis [29].

In multivariable logistic regression models, the effect of sex, age, marital status, presence of COVID-19 symptoms, presence of chronic comorbidity, complications during COVID-19, taking medicine for COVID-19 symptoms, use of the internet during isolation, watch TV during isolation, smoking history and alcohol history was adjusted to identify the factors associated with anxiety symptoms. Similarly for depression, the effect of sex, age, education, family type, occupation, marital status, having health workers in family members, presence of COVID-19 symptoms, complications during COVID-19, taking medicine for COVID-19 symptoms, use of the internet during isolation, watch TV during isolation, get information from WHO/MOHP, and smoking history was adjusted.

## Ethics

Ethical approval for the study was given by the Nepal Health Research Council (825/2020). Informed consent was taken from study participants before the interview and after carefully explaining the study's objectives. As the data was collected through the telephone call, informed consent was taken orally. Personal identifiers such as name were not collected during the study.

## Results

### Prevalence of anxiety and depression

Table 1 shows the level of anxiety and depression among home isolated patients. Out of 372 participants, 74.2% (n = 276) had symptoms of anxiety (borderline: 48.7% and abnormal: 25.5%). Similarly, 79% (n = 294) of the participants experienced symptoms of depression (borderline: 52.7% and abnormal: 26.3%). There was a significant difference in depression status (p = 0.01) across male and female home isolated patients.

**Table 1. Prevalence of anxiety and depression among study participants by gender.**

| Mental health outcomes | Categories | Total (n = 372)<br>n (%) | Male (n = 224)<br>n (%) | Female (n = 148)<br>n (%) | P value |
|---|---|---|---|---|---|
| Anxiety | Normal | 96 (25.8) | 66 (29.5) | 30 (20.3) | 0.1 |
|  | Borderline | 181 (48.7) | 102 (45.5) | 79 (53.4) |  |
|  | Abnormal | 95 (25.5) | 56 (25.0) | 39 (26.4) |  |
| Depression | Normal | 78 (21) | 58 (25.9) | 20 (13.5) | 0.01 |
|  | Borderline | 196 (52.7) | 109 (48.7) | 87 (58.8) |  |
|  | Abnormal | 98 (26.3) | 57 (25.4) | 41 (27.7) |  |

## Sociodemographic characteristics of the study participants

Table 2 shows the association between socio-demographic characteristics of the home isolated COVID-19 patients and their mental health status. There was a significant difference in anxiety (p = 0.04) and depression (p = 0.004) levels across gender with females reporting a higher proportion of anxiety and depression symptoms than males. Marital status was also significantly associated with both anxiety (p = 0.004) and depression (p<0.001). Age (p = 0.004), educational status (p<0.001) and occupation (p = 0.006) was significantly associated with anxiety.

## COVID-19 and behavioral related characteristics of the participants

Table 3 shows the association of anxiety and depression symptoms with COVID-19 related as well as behavioral characteristics of the study participants. The presence of COVID symptoms, complications during COVID-19, taking medicines for COVID-19 symptoms, watching TV during COVID-19, and smoking history was associated with the presence of both anxiety and depression symptoms (p<0.05). Likewise, alcohol history was associated with anxiety while having any chronic morbidity and use of the internet during isolation was associated with depression (p<0.05).

## Factors associated with anxiety and depression among home isolated COVID-19 patients

Watching TV during the home isolation was significantly associated with lowers odds of experiencing symptoms of anxiety (AOR:0.3; 95% CI:0.2–0.7), and depression (AOR:0.4;

**Table 2. Sociodemographic characteristics of the participants.**

| Variables | Anxiety symptoms | | P-value | Depression symptoms | | P-value |
|---|---|---|---|---|---|---|
| | Yes n (%) | No n (%) | | Yes n (%) | No n (%) | |
| **Gender** | | | | | | |
| Male | 158 (70.5) | 66 (29.5) | 0.04 | 166 (74.1) | 58 (25.9) | 0.004 |
| Female | 118 (79.7) | 30 (20.3) | | 128 (86.5) | 20 (13.5) | |
| **Age (in years)** | | | | | | |
| 20–29 | 64 (67.4) | 31 (32.6) | 0.05 | 63 (66.3) | 32 (33.7) | 0.004 |
| 30–39 | 69 (75) | 23 (25.0) | | 76 (82.6) | 16 (17.4) | |
| 40–49 | 55 (69.6) | 24 (30.4) | | 64 (81) | 15 (19.0) | |
| 50 and above | 88 (83) | 18 (17.0) | | 91 (85.8) | 15 (14.2) | |
| **Education** | | | | | | |
| No schooling | 27 (81.8) | 6 (18.2) | | 30 (90.9) | 3 (9.1) | <0.001 |
| Secondary | 89 (78.1) | 25 (21.9) | | 103 (90.4) | 11 (9.6) | |
| Higher secondary | 68 (75.6) | 22 (24.4) | | 67 (74.4) | 23 (25.6) | |
| Graduation or above | 92 (68.1) | 43 (31.9) | 0.2 | 94 (69.6) | 41 (30.4) | |
| **Family type** | | | | | | |
| Nuclear | 167 (72) | 65 (28.0) | 0.2 | 176 (75.9) | 56 (24.1) | 0.05 |
| Joint/Extended | 109 (77.9) | 31 (22.1) | | 118 (84.3) | 22 (15.7) | |
| **Occupation** | | | | | | |
| Employee | 207 (73.9) | 73 (26.1) | 0.8 | 212 (75.7) | 68 (24.3) | 0.006 |
| Unemployed | 69 (75.0) | 23 (25.0) | | 82 (89.1) | 10 (10.9) | |
| **Marital status** | | | | | | |
| Never married | 39 (60.0) | 26 (40.0) | 0.004 | 40 (61.5) | 25 (38.5) | <0.001 |
| Ever married | 237 (77.2) | 70 (22.8) | | 254 (82.7) | 53 (17.3) | |

**Table 3. General COVID-19 related and behavioral characteristics of the participants.**

| Variables | Anxiety symptoms | | P-value | Depression | | P-value |
|---|---|---|---|---|---|---|
| | Yes n (%) | No n (%) | | Yes n (%) | No n (%) | |
| **Have health workers in family members** | | | | | | |
| Yes | 72 (70.6) | 30 (29.4) | 0.3 | 74 (72.5) | 28 (27.5) | 0.06 |
| No | 204 (75.6) | 66 (24.4) | | 220 (81.5) | 50 (18.5) | |
| **COVID Symptoms** | | | | | | |
| Yes | 204 (77.6) | 59 (22.4) | 0.02 | 219 (83.3) | 44 (16.7) | 0.002 |
| No | 72 (66.1) | 37 (33.9) | | 75 (68.8) | 34 (31.2) | |
| **Any chronic comorbidity** | | | | | | |
| Yes | 86 (82.7) | 18 (17.3) | 0.02 | 86 (82.7) | 18 (17.3) | 0.3 |
| No | '90 (70.9) | 78 (29.1) | | 208 (77.6) | 60 (22.4) | |
| **Any complications during COVID-19** | | | | | | |
| Yes | 38 (92.7) | 3 (7.3) | 0.004 | 39 (95.1) | 2 (4.9) | 0.007 |
| No | 239 (71.9) | 93 (28.1) | | 255 (77) | 76 (23) | |
| **Took medicine for COVID-19 symptoms** | | | | | | |
| Yes | 184 (80.3) | 45 (19.7) | 0.001 | 190 (83) | 39 (17) | 0.01 |
| No | 92 (64.3) | 51 (35.7) | | 104 (72.7) | 39 (27.3) | |
| **Use of the internet during isolation** | | | | | | |
| Yes | 230 (72.8) | 86 (27.2) | 0.1 | 242 (76.6) | 74 (23.4) | 0.006 |
| No | 46 (82.1) | 10 (17.9) | | 52 (92.9) | 4 (7.1) | |
| **Watch TV during isolation** | | | | | | |
| Yes | 44 (58.7) | 31 (41.3) | 0.001 | 50 (66.7) | 25 (33.3) | 0.003 |
| No | 232 (78.1) | 65 (21.9) | | 244 (82.2) | 53 (17.8) | |
| **Use of social media for COVID-19 information** | | | | | | |
| Yes | 163 (71.2) | 66 (28.8) | 0.5 | 174 (76) | 55 (24) | 0.7 |
| No | 40 (75.5) | 13 (24.5) | | 39 (73.6) | 14 (26.4) | |
| **Got information from WHO/MOHP** | | | | | | |
| Yes | 109 (74.7) | 37 (25.3) | 0.8 | 108 (74) | 38 (26) | 0.05 |
| No | 167 (73.9) | 59 (26.1) | | 186 (82.3) | 40 (17.7) | |
| **Smoking history** | | | | | | |
| Yes | 17 (58.6) | 12 (41.4) | 0.04 | 18 (62.1) | 11 (37.9) | 0.01 |
| No | 258 (75.7) | 83 (24.3) | | 275 (80.6) | 66 (19.4) | |
| **Alcohol history** | | | | | | |
| Yes | 26 (61.9) | 16 (38.1) | 0.05 | 6 (75) | 2 (25) | 0.7 |
| No | 250 (75.8) | 80 (24.2) | | 286 (79) | 76 (21) | |

95% CI:0.2–0.7). Compared to never married, ever married study participants had significantly higher odds of exhibiting symptoms of anxiety (AOR: 2.9; 95% CI:1.3–6.6). Participants who had COVID-19 complications during home isolation (AOR: 3.7; 95% CI:1.1–10.1) and those who used medicine for COVID-19 symptoms (AOR:1.9; 95% CI:1.1–3.5) had a significantly higher likelihood of experiencing anxiety symptoms compared to those not having complications and those not using the medicine for COVID-19 respectively.

Regarding gender, males (AOR: 0.4; 95% CI: 0.2–0.8) had significantly lower odds of having symptoms of depression than females. Participants who developed COVID-19 symptoms during home isolation (AOR:3.3; 95% CI:1.6–6.7) had higher odds of having depression compared to those who did not have COVID-19 symptoms. However, age, family type, occupation,

**Table 4. Factors associated with anxiety and depressive symptoms.**

| Variables | Anxiety | | Depression | |
|---|---|---|---|---|
| | Crude odds ratio 95% CI | Adjusted odds ratio 95% CI | Crude odds ratio 95% CI | Adjusted odds ratio 95% CI |
| **Sex (Ref: Female)** | | | | |
| Male | 0.6 (0.4–0.9)* | 0.7 (0.4–1.1) | 0.4 (0.3–0.7)** | **0.4 (0.2–0.8)*** |
| **Age in years (Ref: 50 and above)** | | | | |
| 20–29 | 0.4 (0.2–0.8)* | 0.9 (0.4–2.5) | 0.3 (0.2–0.6)* | 0.8 (0.3–2.1) |
| 30–39 | 0.6 (0.3–1.2) | 0.7 (0.3–1.5) | 0.8 (0.3–1.7) | 1.1 (0.4–2.6) |
| 40–49 | 0.5 (0.2–0.9)* | 0.5 (0.2–1.1) | 0.7 (0.3–1.5) | 0.8 (0.3–2.0) |
| **Education (Ref: No schooling)** | | | | |
| Graduation or above | | | 0.3 (0.1–0.8)* | 0.4 (0.1–1.6) |
| Higher secondary | | | 0.3 (0.1–1.0) | 0.4 (0.1–1.7) |
| Secondary | | | 0.9 (0.2–3.5) | 1.3 (0.3–5.9) |
| **Family type (Ref: Nuclear)** | | | | |
| Joint/Extended | | | 0.6 (0.3–1.0) | 0.6 (0.3–1.2) |
| **Occupation (Ref: Employed)** | | | | |
| Unemployed | | | 2.6 (1.3–5.4)** | 1.0 (0.4–2.5) |
| **Marital status (Ref: Never married)** | | | | |
| Ever married | 2.3 (1.3–3.9)* | **2.9 (1.3–6.6)*** | 3.0 (1.7–5.4)*** | 2 (0.9–5.1) |
| **Have health workers in family members (Ref: No)** | | | | |
| Yes | | | 0.6 (0.4–1.0) | 0.8 (0.4–1.6) |
| **COVID Symptoms (Ref: No)** | | | | |
| Yes | 1.7 (1.1–2.9) | 1.6 (0.9–3.1) | 2.3 (1.3–3.8)** | **3.3 (1.6–6.7)*** |
| **Any chronic comorbidity (Ref: No)** | | | | |
| Yes | 1.9 (1.1–3.4)* | 1.6 (0.8–3.2) | | |
| **Any complications during COVID-19 (Ref: No)** | | | | |
| Yes | 4.9 (1.5–16.4)** | **3.7 (1.1–10.1)*** | 2.3 (1.3–3.8)** | 3.6 (0.8–16.2) |
| **Took medicine for COVID-19 symptoms (Ref: No)** | | | | |
| Yes | 1.8 (1.1–2.9)* | **1.9 (1.1–3.5)*** | 1.8 (1.1–3.0)* | 1.4 (0.7–2.6) |
| **Use of internet during isolation** | | | | |
| Yes | 0.6 (0.3–1.2) | 0.6 (0.3–1.3) | 0.3 (0.1–0.7)* | 0.4 (0.1–1.1) |
| **Watch TV during isolation (Ref: No)** | | | | |
| Yes | 0.4 (0.2–0.7)** | **0.3 (0.2–0.6)*** | 0.4 (0.2–0.8)* | **0.4 (0.2–0.7)*** |
| **Got information from WHO/MOHP (Ref: No)** | | | | |
| Yes | | | 0.6 (0.3–1.0) | 0.9 (0.5–1.8) |
| **Smoking history (Ref: Yes)** | | | | |
| No | 2.2 (1.0–4.8)* | 1.3 (0.5–3.3) | 2.5 (1.1–5.6)* | 1.8 (0.7–5.0) |
| **Alcohol history (Ref: Yes)** | | | | |
| No | 1.9 (1.0–3.7) | 1.7 (0.8–3.2) | | |

*p-value less than 0.05

**p-value less than 0.01

***p-value less than 0.001.

education, having health workers in family members, presence of chronic comorbidities, use of the internet during home isolation, smoking history, alcohol history, and use of the MOHP/WHO website for the information was not significantly associated with the symptoms of anxiety and depression (Table 4).

## Discussion

To the best of our knowledge, this is the first study to identify the prevalence of anxiety and depressive symptoms among the home isolated COVID-19 patients and its associated factors in Nepal. The prevalence of anxiety and depressive symptoms was found in the majority of the home isolated patients with more than half having borderline symptoms and one in four having abnormal symptoms. This prevalence rate was higher than that reported in a recent meta-analysis which showed the prevalence of anxiety and depressive symptoms among patients with COVID-19 were 47% and 45%, respectively [30]. Compared to the previous studies in developing countries among COVID-19 patients in isolation, the prevalence of anxiety symptoms in the present study was higher than study conducted in Iran; 29.3% [31] and Wuhan, China; 18.6% [32] while lower than study conducted in Iran 100% [33]. The variation among countries could be due to different health system responses, sociodemographic compositions and different psychometric instruments used to measure anxiety and depression. Importantly, the study findings reinforce that the mental health burden during the pandemic should not be neglected, and thus mental health services should be a core part of the COVID-19 response plan. Considering that the mental health resources are scarce at primary health care level [34] and the pandemic saw an increase in suicide rate [35], we suggest for the deployment of trained mid-level health workers and psychological counsellors for offering consultation as well as linkage to hospitals and centers with treatment facilities.

Our study findings indicated that females had higher odds of having depression symptoms as compared to males. This finding is in line with the result of previous studies conducted in Italy [36], China [37], Spain [38] and Bangladesh [39]. These differences could be due to greater exposure to stresses during the pandemic and/or heightened response to stress in females. As females in Nepal are usually the caregivers in a family and are involved in domestic works, infection could further traumatize them for not being able to perform their role Likewise, females are more prone to develop internalizing symptoms following exposure to stress and trauma, even accounting for the specific event [40].

The present study found that COVID-19 home isolated patients who watched television during isolation were more likely to have low anxiety and depressive symptoms than those who did not watch television. This result contradicts the Italian study, which showed that people who spent more time watching TV series during the pandemic lockdown reported higher levels of anxiety [41]. A study from Netherland also found that computer use and television viewing were linked to anxiety and/or depressive symptoms [42]. Exposure to social media and thinking about the pandemic for a long time could have worse psychological consequences [43,44]. In our study participants, watching television could have provided an opportunity to avert patients from overthinking and thus possible mental health problems. However, we did not measure the content and intensity of exposure to television and thus it is only the author's assumption. Although not significant, those using internet also had lower odds of experiencing depression symptoms. Further studies could investigate the association of mental health symptoms with exposure to content among those watching mass media and social media.

The current study showed that the relationship status can contribute to anxiety. Married people had greater odds of anxiety symptoms during home isolation. This can stem from dissatisfaction or the perception of a support imbalance [45]. Married persons are more likely to think about their family members and have more incumbents which might have resulted in having higher anxiety and depression. On other hand, never married person enjoy their freedom and have carefree lifestyles.

Likewise, participants who had COVID-19 complications during home isolation had higher odds of having anxiety as compared with participants who did not have any complications.

This finding is in line with the previous study conducted in Cameroon [46]. The observed association of high levels of anxiety with COVID-19 complications has been demonstrated in a study by Mazza et al [47]. In their study, it was postulated that increased levels of anxiety as a long-term sequela of COVID-19 infection could be explained by the inflammatory changes caused by the infection.

Our analysis revealed that having COVID-19-related symptoms was associated with depressive symptoms among COVID-19 home isolation patients. This finding was supported by a similar study conducted in Bangladesh which reported that having COVID-19-related physical symptoms was associated with depressive symptoms among inpatients in COVID-19 isolation facilities. This could be explained because COVID-19 symptoms like fever, shortness of breath, and headache can produce mental effects among patients [48]. Studies elsewhere also have shown that psychological distress symptoms such as anxiety and depression are common in patients with more clinical symptoms and illness severity [49,50]. Patients with clinical symptoms could also be more worried about the prognosis of the disease.

## Study limitation

There are some limitations of this study, which need to be acknowledged. This study was conducted during the early phase of the pandemic when treatment and vaccines were not available and thus could have affected the presence of anxiety and depression symptoms. Similarly, mental health outcomes might still reflect conditions existing before this pandemic. Besides, the present study lacked clinical interviews to confirm the diagnosis of anxiety and depression and the study findings may not be generalizable to study participants from other provinces of Nepal. Also, we have not included the history of mental illness and medications taken for any kind of mental illness before the pandemic. Despite these limitations, this study provides evidence on mental health status among home COVID-19 isolated patients. To the best of our knowledge, this is the first study in Nepal and could aid to scarce evidence available regarding the mental health status among home isolated COVID-19 patients. This evidence could be of interest to policymakers, and various stakeholders who are involved in the response to COVID-19 or any future epidemic.

## Conclusion

In summary, the findings have shown that a substantial proportion of COVID-19 patients in Nepal experienced depressive and anxiety symptoms during home isolation with more than half having borderline and one out of four having abnormal mental health symptoms. Female and those with COVID-19 symptoms had higher odds of exhibiting depression symptoms while ever married, those with COVID-19 related complications and those who took medicines for treatment of symptoms were at higher odds of developing anxiety symptoms. Interestingly, watching Television during isolation was associated with lower odds of developing anxiety and depression symptoms. Considering the burden of mental health symptoms during the pandemic, we urge the Government of Nepal for mobilizing mental health human resources such as psychiatrists, psychologists, psychiatric nurses, and trained health workers and community mobilizers with appropriate linkage to treatment services. We suggest that more attention and timely psychological interventions be given to the COVID-19 home isolated patients with connection to mental health services.

## Supporting information

**S1 Appendix. STROBE statement checklist.**
(DOC)

## Acknowledgments

The authors would like to acknowledge enumerators and all the study participants for their participation in the survey. The authors are grateful to Ministry of Health, Province One for their administrative support.

## Author Contributions

**Conceptualization:** Pratik Khanal, Kiran Paudel, Suresh Mehata, Astha Thapa, Hari Krishna Bhattarai.

**Data curation:** Pratik Khanal, Hari Krishna Bhattarai.

**Formal analysis:** Pratik Khanal, Kiran Paudel, Ramesh Bhatta.

**Funding acquisition:** Hari Krishna Bhattarai.

**Investigation:** Pratik Khanal, Kiran Paudel, Hari Krishna Bhattarai.

**Methodology:** Pratik Khanal, Kiran Paudel, Hari Krishna Bhattarai.

**Project administration:** Suresh Mehata, Astha Thapa, Hari Krishna Bhattarai.

**Resources:** Hari Krishna Bhattarai.

**Supervision:** Suresh Mehata, Astha Thapa, Hari Krishna Bhattarai.

**Validation:** Pratik Khanal, Kiran Paudel.

**Visualization:** Pratik Khanal, Kiran Paudel, Ramesh Bhatta.

**Writing – original draft:** Pratik Khanal, Kiran Paudel, Ramesh Bhatta.

**Writing – review & editing:** Suresh Mehata, Astha Thapa, Hari Krishna Bhattarai.

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
