## [Decision Letter · Decision Letter 0]

20 Jul 2022

PGPH-D-22-00768

Anxiety and depressive symptoms among home isolated patients with COVID-19 in a low-resource setting: a cross-sectional study from Nepal

Dear Dr.  Khanal,

Thank you for submitting your manuscript to PLOS Global Public Health. After careful consideration, we feel that it has merit but does not fully meet PLOS Global Public Health’s publication criteria as it currently stands. Therefore, we invite you to submit a revised version of the manuscript that addresses the points raised during the review process.

We look forward to receiving your revised manuscript.

Kind regards,

Peter Bai James, PhD

Academic Editor

Journal Requirements:

1. Please amend your online Financial Disclosure statement. If you did not receive any funding for this study, please simply state: “The authors received no specific funding for this work.”

2. Please update your online Competing Interests statement. If you have no competing interests to declare, please state: “The authors have declared that no competing interests exist.”

Additional Editor Comments (if provided):

Please provide a supplementary file to show that the current study adheres to STROBE guidelines for cross-sectional study. 

Please narrow your conclusion to the area of study. Avoid generalizing your findings for the whole of Nepal as it may not hold true given that only one area of Nepal was considered in your study.

Reviewers' comments:

Reviewer's Responses to Questions

**Comments to the Author**

1. Does this manuscript meet PLOS Global Public Health’s publication criteria? Is the manuscript technically sound, and do the data support the conclusions? The manuscript must describe methodologically and ethically rigorous research with conclusions that are appropriately drawn based on the data presented.

Reviewer #1: Yes

Reviewer #2: Yes

2. Has the statistical analysis been performed appropriately and rigorously?

Reviewer #1: Yes

Reviewer #2: Yes

3. Have the authors made all data underlying the findings in their manuscript fully available (please refer to the Data Availability Statement at the start of the manuscript PDF file)?

Reviewer #1: Yes

Reviewer #2: Yes

4. Is the manuscript presented in an intelligible fashion and written in standard English?

Reviewer #1: Yes

Reviewer #2: Yes

5. Review Comments to the Author

Reviewer #1: Congratulations on completing your work and disseminating the results. This manuscript relevant to the current situation in the midst of the Covid-19 pandemic. I have some feedback intended to be helpful, and I wish you the best of luck in your ongoing research. Take care.

Introduction

- I found that the subject matter was important, and the knowledge gap is presented in the introduction.

- A small but relatively significant comparison revealed that the purpose as outlined in the abstract and the body of the paper are different (please make this consistent).

Method

- Regarding “COVID-19 patients who had 96 spent at least 5 days at home isolation”, please put the rationale for this approach.

- Please kindly put the information about psychometric properties such as validity and reliability all of the instruments.

Discussion

- If possible, please add an alternative explanation regarding study finding about female participant had higher odds from cultural perspective in your country.

Conclusion

-Some of your conclusion is not coherent with your findings, please make sure answer the research question in your conclusion.

Reviewer #2: This is an important piece of work, demonstrating the mental health burden among individuals living in home isolation after testing positive for COVID-19 in Nepal. Insofar as I wish that researchers could have expanded the geography to include other parts of the country and compared the mental health burden between home-isolated versus the general population, I understand the challenges in obtaining primary data for the study with a broader scope. Still, I find this research useful, and within the scope of this journal.

The study is generally well-presented, is relatively straightforward, and has used an easy but established methodology. Below are some of my comments, many of them being minor. I would be happy to take a revised version of this manuscript.

Title: I can understand the authors’ temptation to highlight “low resource settings” for this story conducted in Nepal but I note that this study is neither nationally representative nor conducted in the most impoverished geography within Nepal (In fact, province one is among the one with better HDI). Moreover, the poor-resource setting is not much discussed in the manuscript text. Therefore, I would recommend dropping the word “low resource settings” from the title.

Abstract: I think it makes sense to move up the sentence starting in line 42. This way, the abstract reads more coherent: first, you highlight poor mental health is a big issue and then you study factors contributing to it. This is also the order results are presented in the main text. In line 34, I would recommend being more specific and replacing “multivariable analysis” with “multivariable logistic regression analysis”. Also, in line 34, to the casual readers, it’s confusing what 0-21 means. So, just remove “(HADS:0-21)”.

In Line 44: It is not clear what “connection to the health system” means. I think that the significance of this result holds whether or not the home isolations are tied to government-imposed forced quarantines or self-quarantine. If authors want to restrict their inference to a particular type of quarantine scheme, please make this sentence a bit clearer.

Keywords: Mental Health could be another potential keyword here.

Introduction: This section can be tightened in a few places. For example, for this study, it is more important to highlight when the Covid case/death was first reported in Nepal and Province 1, rather than pretty-much-known info about Wuhan.

On line 62: Please re-phrase the sentence “The onslaught of ….” so that this statement looks relevant years from now.

Line 71: This sentence “A study from Saudi Arabia reported 17.1%...” reads awkwardly. Rephrase this one, emphasizing that these numbers are prevalence estimates.

On line 78: white space in “ofNepal”

On line 82: I don’t think “close supervision of local health workers” is accurate. It was one of the haphazard, ad-doc decisions mostly carried out by the local government. I know you cited a gov report but they should be taken with a grain of salt.

Line 86: I would not use “could have been” because it usually means something was possible in the past but did not happen.

Line 87: change “During” to something like “As of”. I get what you trying to say here but reads a little off.

Methods:

Line 95: I am not a clinician, but I’d probably not use the term “patients” here and elsewhere since these are the individuals who tested positive for the virus (and not necessarily showing symptoms of the disease).

Line 96: It might be helpful for international readers to add a little context to the province you are studying.

Line 99: white space in “2021and”

Line 104: Missing hyphen in “RT PCR”

In the sample size calculation, a general comment: it is important to note that you are estimating the minimum sample size required to draw a reasonable inference. It is always preferable to use more information when possible. In other words, the number you obtained by plugging your assumptions (i.e. 372) is not a magic number that can’t be exceeded in selecting respondents, and when possible, aim to recruit more participants.

Line 111: white space in “of372”

Line 121: “which are scored from 0 to 21”: Either mention that it’s the “total” score that ranges from 0 to 21, or write that each individual question is scored from 0 to 3.

Line 125: Please try to rephrase “study tools” in this context.

Line 127: comma not needed in (Nuclear, and Joint/Extended)

Line 137: Chi-square test does not distinguish “independent” or “dependent” variables. Just say categorical variables.

Double check whether you adjusted for chronic comorbidities for both anxiety and depression. In the methods, you say this goes into models for both outcomes but the result table says otherwise.

Results: The exceptionally high prevalence, particularly for depression, made me double-think about the validity of the instrument used to identify these outcomes and the willingness/understanding/wording during the interview process. A vast majority have a borderline problem, perhaps indicating central tendency bias of respondents. Assuming that people with borderline health problems could be cared for with little resources than those with chronic/serious MH patients, it suggests that even the smaller efforts put into mental health could go a long way. I think that this could be something authors can consider discussing and calling for action in their discussion/conclusion.

Table 4. One way to reduce the unnecessary rows is to specify the reference category when you mention the variable name. It makes the table easier to read and avoids countless Ref/Ref.

I see no value in displaying crude ORs. These can be easily derived from the characteristics of the respondent’s tables displayed previously. But it’s up to the authors whether to keep it or not.

Discussion: The authors do a fairly good job in this section, but I wanted to see a more nuanced and focused discussion around the mental health context in Nepal and what authorities and researchers should do next. For example, please put the research in the broader context of mental health problems—Nepal is one of the few countries that saw an increase in the suicide rate during the pandemic. (https://journals.plos.org/plosone/article?id=10.1371/journal.pone.0262958). This research can be easily molded to the existing body of knowledge from within Nepal (I don’t see that paper in the reference). Given the worrying mental health prevalence in province 1, how important is it to take this issue seriously in other provinces that are even poorer? What proportions of respondents were in imposed quarantine facilities or how many were in their own homes? Would the elevated mental health problems among women might indicate the forced quarantines were not women-friendly? Maybe authors might comment something on TV use and better mental health?

6. PLOS authors have the option to publish the peer review history of their article (what does this mean?). If published, this will include your full peer review and any attached files.

**Do you want your identity to be public for this peer review?** For information about this choice, including consent withdrawal, please see our Privacy Policy.

Reviewer #1: No

Reviewer #2: No

---

## [Decision Letter · Decision Letter 1]

9 Aug 2022

PGPH-D-22-00768R1

Anxiety and depressive symptoms among home isolated patients with COVID-19: a cross-sectional study from Province One, Nepal

Dear Dr. Khanal,

Thank you for submitting your manuscript to PLOS Global Public Health. After careful consideration, we feel that it has merit but does not fully meet PLOS Global Public Health’s publication criteria as it currently stands. Therefore, we invite you to submit a revised version of the manuscript that addresses the points raised during the review process.

We look forward to receiving your revised manuscript.

Kind regards,

Peter Bai James, PhD

Academic Editor

Journal Requirements:

Additional Editor Comments (if provided):

Please add a supplementary file indicating your study adheres to STROBE guideline for cross-sectional studies. 

You can access a copy of the guideline can be accessed via this link Best Practices in Research Reporting | PLOS Global Public Health

Reviewers' comments:

Reviewer's Responses to Questions

**Comments to the Author**

1. If the authors have adequately addressed your comments raised in a previous round of review and you feel that this manuscript is now acceptable for publication, you may indicate that here to bypass the “Comments to the Author” section, enter your conflict of interest statement in the “Confidential to Editor” section, and submit your "Accept" recommendation.

Reviewer #1: All comments have been addressed

Reviewer #2: All comments have been addressed

2. Does this manuscript meet PLOS Global Public Health’s publication criteria? Is the manuscript technically sound, and do the data support the conclusions? The manuscript must describe methodologically and ethically rigorous research with conclusions that are appropriately drawn based on the data presented.

Reviewer #1: Yes

Reviewer #2: Yes

3. Has the statistical analysis been performed appropriately and rigorously?

Reviewer #1: Yes

Reviewer #2: Yes

4. Have the authors made all data underlying the findings in their manuscript fully available (please refer to the Data Availability Statement at the start of the manuscript PDF file)?

Reviewer #1: Yes

Reviewer #2: Yes

5. Is the manuscript presented in an intelligible fashion and written in standard English?

Reviewer #1: Yes

Reviewer #2: Yes

6. Review Comments to the Author

Reviewer #1: The authors have done an excellent job responding to reviewer feedback, both mine and that of the other reviewers. I was able identify my comments and how they were addressed, as well as that of the others. I have no follow-up recommendations and suggest accept without further revision. Congratulations on an excellent response.

Reviewer #2: Thank you for addressing all the concerns and thorough revision. I look forward for the positive editorial decision in this important piece of work.

7. PLOS authors have the option to publish the peer review history of their article (what does this mean?). If published, this will include your full peer review and any attached files.

**Do you want your identity to be public for this peer review?** For information about this choice, including consent withdrawal, please see our Privacy Policy.

Reviewer #1: No

Reviewer #2: No

---

## [Editor Report · Decision Letter 2]

19 Aug 2022

Anxiety and depressive symptoms among home isolated patients with COVID-19: a cross-sectional study from Province One, Nepal

PGPH-D-22-00768R2

Dear Dr Khanal,

We are pleased to inform you that your manuscript 'Anxiety and depressive symptoms among home isolated patients with COVID-19: a cross-sectional study from Province One, Nepal' has been provisionally accepted for publication in PLOS Global Public Health.

Best regards,

Peter Bai James, PhD

Academic Editor
